# Exosomes in the Tumor Microenvironment: From Biology to Clinical Applications

**DOI:** 10.3390/cells10102617

**Published:** 2021-10-01

**Authors:** Vitor Rodrigues da Costa, Rodrigo Pinheiro Araldi, Hugo Vigerelli, Fernanda D’Ámelio, Thais Biude Mendes, Vivian Gonzaga, Bruna Policíquio, Gabriel Avelar Colozza-Gama, Cristiane Wenceslau Valverde, Irina Kerkis

**Affiliations:** 1Programa de Pós-Graduação em Biologia Estrutural e Funcional, Escola Paulista de Medicina (EPM), Federal University of São Paulo (UNIFES), São Paulo 04039-032, Brazil; vitor_rodriguesdacosta@hotmail.com (V.R.d.C.); thais.biude.mendes@gmail.com (T.B.M.); avelarbio46@gmail.com (G.A.C.-G.); 2Genetics Laboratory, Instituto Butantan, São Paulo 05508-010, Brazil; hugo.barros@esib.butantan.gov.br (H.V.); fernanda.damelio@esib.butantan.gov.br (F.D.); vivia_gonzaga@hotmail.com (V.G.); bruna.policiquio@gmail.com (B.P.); 3Cellavita Pesquisas Científicas Ltd.a., Valinhos 13271-650, Brazil; cristiane.valverde@cellavitabrasil.com.br; 4Genetic Bases of Thyroid Tumors Laboratory, Division of Genetics, Department of Morphology and Genetics, Federal University of São Paulo (UNIFESP), São Paulo 04039-032, Brazil

**Keywords:** exosomes, cancer, tumor microenvironment (TME), immunomodulation, epithelial-mesenchymal transition (EMT), mesenchymal-stem cell (MSC), cell-free therapy

## Abstract

Cancer is one of the most important health problems and the second leading cause of death worldwide. Despite the advances in oncology, cancer heterogeneity remains challenging to therapeutics. This is because the exosome-mediated crosstalk between cancer and non-cancer cells within the tumor microenvironment (TME) contributes to the acquisition of all hallmarks of cancer and leads to the formation of cancer stem cells (CSCs), which exhibit resistance to a range of anticancer drugs. Thus, this review aims to summarize the role of TME-derived exosomes in cancer biology and explore the clinical potential of mesenchymal stem-cell-derived exosomes as a cancer treatment, discussing future prospects of cell-free therapy for cancer treatment and challenges to be overcome.

## 1. Exosomes Mediate Crosstalk between Cancer and Non-Cancer Cells within the Tumor Microenvironment

Cancer is one of the most important health problems worldwide and the second leading cause of death globally [1]. According to GLOBOCAN, nearly 19.3 million new cancer cases and almost 10 million cancer deaths occurred in 2020 [1]. However, cancer incidence and mortality are rapidly growing worldwide due to population aging and growth [1,2]. Based on the statistical projections, the International Agency for Research on Cancer (IARC) estimates that more than 28.8 million new cancer cases and 16.1 million cancer deaths will occur in 2040 [3].

Despite the advances in molecular oncology that have driven the identification of tumor genotype variations between patients (interpatient heterogeneity), the presence of subpopulations of cancer cells with unique genomes in the same patient (intratumor heterogeneity) presents challenges to cancer therapeutics [4,5]. In this sense, cumulative evidence has shown that cancer cells communicate with different populations of non-cancer cells within the tumor microenvironment (TME) [6]. This communication is mediated by a plethora of bioactive molecules, including proteins, lipids, coding and non-coding RNAs, and metabolites, which are secreted into nanosized vesicles known as exosomes (30–200 nm) [7,8].

On the one hand, the discovery of the role of these exosomes in cancer biology has allowed us to understand the complexity of the TME; on the other hand, it has also allowed us to explore the biotechnological potential of mesenchymal stem cell (MSC)-derived exosomes as therapeutics for cancer treatment in a novel therapeutic approach known as cell-free therapy. Based on the recent discoveries in exosome-related cancer biology and biotechnology, this review aims to summarize the role of these vesicles in all carcinogenesis steps and highlight the clinical applications of MSC-derived exosomes for cancer treatment, discussing the future prospects of cell-free therapy in the oncology field.

## 2. Exosome Biogenesis

Naturally, all cell types produce and secrete different types of extracellular vesicles (EVs), which participate in both physiological and pathophysiological processes [9,10]. Depending on their size, biogenesis mechanisms, or function, these vesicles are classified as microvesicles (100–1000 nm), exosomes (30–200 nm), or apoptotic bodies (generally > 1000 nm) [11,12,13].

Typically, exosomes are surrounded by a phospholipid membrane containing an abundance of cholesterol, sphingomyelin, ceramide, lipid rafts, and evolutionarily conserved biomarkers, which are used to distinguish them from microvesicles or apoptotic bodies, such as tetraspanins (CD9, CD63, CD81, and CD82), heat shock proteins (Hsp60, 70, and 90), major histocompatibility component classes I (MHC-I) and II (MHC-II), Alix, Tsg101, lactadherin, and lysosome-associated membrane glycoprotein 2, as illustrated in Figure 1 [11,14,15,16,17,18]. Besides these proteins, exosomes contain specific proteins and transcripts, which are responsible for eliciting the regulation of recipient cells.

Exosomes were discovered in 1983 [19,20,21]. However, they were initially proposed as cellular waste resulting from cell damage or by-products of cell homeostasis [20,22]. Since their discovery, it has become clear that these vesicles act as a key mediator of cell-to-cell communication [22,23].

Exosomes are generated from late endosomes, formed by inward budding of the early endosomes, which later mature into multivesicular bodies (MVBs) [18,24]. Invagination of late endosomal membranes results in the formation of ILVs within MVBs [22,25]. Certain proteins are incorporated into the membrane’s invagination during this process, while the cytosolic components are engulfed and enclosed within the ILVs [22].

Upon maturation, MVBs destined for exocytosis are transported to the plasma membrane along microtubules by the Rab GTPases (Rab2b, Rab5a, Rab9a, Rab11, Rab27a, Rab 27b, and Rab35) [26,27,28,29]. After transport to and docking in the plasma membrane, secretory MVBs couple to the soluble N-ethylmaleimide-sensitive component attachment protein receptor (SNARE) membrane fusion machinery [18,26]. Finally, MVBs fuse with the plasma membrane, releasing ILVs into the extracellular space called “exosomes” [18,22].

Secreted exosomes can bind to a neighboring cell, interact with the extracellular matrix (ECM), or passively be transported through the bloodstream and other body fluids, regulating distant recipient cells [13,18,26,30]. However, the in vivo half-life of exosomes is very short in circulation, and up to 90% of exosomes are removed within 5 min [31].

Numerous factors determine the biodistribution of isolated exosomes after their in vivo administration, such as original cells, route of delivery, and targeting condition. The recipient cells absorb exosomes by membrane fusion, endocytosis, or receptor-mediated internalization through a caveolin-, clathrin-, or lipid-raft-mediated phagocytosis or endocytosis mechanism [13,18,22,26,30].

Alternatively, exosomes can interact with the parental cells, resulting in autocrine signaling. Finally, an alternative fate for MVBs is fusion with lysosomes, which leads to degradation and recycling of their protein, nucleotide, and lipid components [13,18,22,26].

Although what distinguishes the MVBs to be secreted from those that will be degraded remains unclear, it is known that the fate of MVBs can change in response to cellular conditions [26]. For example, MVBs are degraded by fusion with autophagosomes under starvation conditions, resulting in decreased exosome release [18,26,32]. The exosome biogenesis process is summarized in Figure 2.

## 3. Molecular Cargo

Exosomes contain selective repertoires of proteins, nucleic acids (RNAs), lipids, and metabolites that regulate signaling pathways in the recipient cells [33]. The enrichment of a particular set of molecules within the exosomes suggests the existence of specific sorting mechanisms that orchestrate the selective packaging of the RNAs and proteins [33].

For many years, the sorting mechanism remained unclear. However, nowadays, it is clear that the selective packaging of RNAs and proteins is governed by the endosomal sorting complex required for transport (ESCRT), which also contributes to exosome formation.

The ESCRT is protein machinery composed of four ESCRT proteins (ESCRT-0, -I, -II, and -III) that work cooperatively to facilitate MVB formation, vesicle budding, and protein cargo sorting [22,34].

The ESCRT-mediated sorting is initiated by recognition and sequestration of ubiquitinated proteins to specific domains (the Hrs FYVE domain with phosphatidylinositol 3-phosphate (PtdIns3P)) of the endosomal membrane via ubiquitin-binding subunits of ESCRT-0 [22,35]. Next, the Hrs PSAP domain of the ESCRT-0 interacts with the subunit tumor susceptibility gene 101 (tsg101) of ESCRT-I [22,35]. ESCRT-I recruits the ESCRT-II proteins, which recruit and activate the ESCRT-III complex, which promotes the budding processes [22,35]. This occurs because the Snf7 protein of the ESCRT-III complex forms oligomeric assemblies, promoting vesicle budding [22,35]. Snf7 also recruits the Alix protein, stabilizing the ESCRT-III assembly [22,35]. Following cleaving the buds to form ILVs, the ESCRT-III complex separates from the MVB membrane with energy supplied by the sorting protein ATP Vps4 [22].

Although ESCRT-III is considered to be required for the scission of the ILVs into the MVE lumen [36], studies have reported the presence of ILVs within the lumen of MVBs in the ESCRT-depleted cells, indicating that ESCRT-independent pathways for ILV formation exist [37,38].

In this sense, recent evidence supports an alternative pathway for sorting exosomal cargo into MVBs in an ESCRT-independent manner, which seems to depend on raft-based microdomains for the lateral segregation of cargo within the endosomal membrane [22,37]. These microdomains are highly enriched in sphingomyelinases, from which ceramides can be formed by hydrolytic removal of the phosphocholine moiety [22,39].

The cone-shaped structure of ceramides might cause spontaneous negative curvature of the endosomal membrane, thereby promoting domain-induced budding [22,39].

In addition, proteins such as tetraspanins also participate in exosome biogenesis and protein loading. Tetraspanin-enriched microdomains (TEMs) are ubiquitous specialized membrane platforms for compartmentalizing receptors and signaling proteins in the plasma membrane [22,40,41].

Thus, by exhibiting sorting mechanisms, which select the proteins and RNAs that will compose the exosome content, it is expected that exosomes derived from non-cancer cells and cancer cells possess distinct activities in both physiology and pathophysiology.

## 4. Cancer-Derived Exosomes in Carcinogenesis

Cells of different tissue types produce and release exosomes to facilitate intercellular communication [24]. For this reason, it is not surprising that cancer-derived exosomes mediate the communication between cancer cells and non-cancer cells within the TME as well as malignant and non-malignant cells, regulating all carcinogenesis steps [42].

Typically, exosomes derived from cancer cells are larger than those derived from non-cancer cells. This size difference can be attributed to the heterogeneous nature of cancer cells, since different subclones of cancer cells are present within the TME, as well as the overexpression of genes related to the carcinogenic process [43]. For this reason, exosomes derived from cancer cells have been referred to as oncosomes (100–400 nm) or large oncosomes (LOs, 1–10 μm) according to their size and cargoes, as illustrated in Figure 3 [43]. Oncosomes are vesicles carrying abnormal and transforming macromolecules, including oncoproteins [43,44]. LOs are atypical extracellular vesicles, produced as a byproduct of non-apoptotic plasma membrane blebbing from cancer cells, and induced by silencing of the cytoskeletal regulator Diaphanous-related formin-3 (DIAPH3), by overexpression of the oncoproteins MyrAKT1, HB-EGF, and caveolin-1, or by the activation of the EGFR [45,46]. However, at present, there is no unanimous consensus on the nomenclature of these extracellular vesicles secreted by cancer cells. Thus, to avoid misinterpretation, herein, we adopt the term “cancer-derived exosomes” to summarize large exosomes and/or oncosomes derived from cancer cells and the term “exosome” to refer to typical exosomes (30–200 nm) secreted by non-cancer cells.

Based on the cumulative evidence supporting the view that these cancer-derived exosomes contribute to all carcinogenesis steps [26,47,48,49,50], this review aims to summarize the role of cancer-derived exosomes in cancer initiation, promotion, progression, and metastasis, highlighting mechanisms of action commonly reported in different malignancies.

### 4.1. Cancer-Derived Exosomes Mediate Crosstalk between Inflammation and Cancer Initiation

Cancer initiation is characterized by irreversible genetic alterations (driver mutation) that lead to the gain of function of oncogenes and/or loss of tumor suppression genes [51]. In addition, these mutations, associated with mitogenic stimuli from pre-cancerous micromilieu (cancer promotion), induce “initiated” cell proliferation (cancer progression). These combined steps increase the genomic instability, facilitating the novel mutations during the somatic evolution (passenger mutation) [52].

Current studies have demonstrated that exosomes are a key mediator of intercellular communication between cancer cells and non-cancer cells within the TME, acting as initiators of carcinogenesis by mediating crosstalk between inflammation and cancer initiation [30,53,54].

Both historically and contemporarily, cancer has been seen as an inflammatory disease [55,56]. However, in the last couple of decades, the contribution of the immune system and inflammation to cancer development has gained an enormous amount of interest [56]. This interest has allowed us to confirm that inflammation predisposes to the development of cancer and contributes to the acquisition of many hallmarks of cancer [56,57,58,59].

In this sense, studies have shown that exosomes produced and released by cancer cells contain various biomolecules, including nuclear factor kappa B (NFκB) and signal transducer activator of transcriptions 3 (STAT3), as well as inflammatory cytokines, such as interleukin (1L)-1β, -6, and tumor necrosis factor-alpha (TNF-α), which promotes the recruitment of immune cells to target sites as revisited by Othman et al. [50].

In 2013, Bretz et al. [60] showed that exosomes obtained from malignant ascites of ovarian cancer patients were able to bind to Toll-like receptors (TLR2 and TLR4) present on the surface of THP-1 cells (a spontaneously immortalized human monocyte-like cell line), inducing the production and secretion of the pro-inflammatory cytokines IL-1β, IL-6, IL-8, and TNF-α in a NFκB- and STAT3-dependent manner.

However, the cancer-derived exosomes’ action is not limited to monocyte recruitment. Studies already demonstrated that breast [61] and gastric cancer-derived exosomes induce the differentiation of monocytes into M1 macrophages in a NFκB-dependent manner, stimulating the production of pro-inflammatory cytokines (GCSF, IL-6, IL-8, IL-1β, CCL2, and TNF-α) [62]. Interestingly, Chow et al. [61] revealed that the activation of NFκB in monocytes/macrophages occurs through cancer-derived exosomes binding to TLR2, emphasizing the Toll-like receptors’ role in the crosstalk between inflammation and cancer initiation and progression.

The release of pro-inflammatory cytokines within the TME also recruits neutrophils (the most abundant leukocytes in the immune system) to the TME [57], leading to the generation of reactive oxygen species (ROS) [59,63]. The oxidative stress can lead to single and/or double-strand DNA breaks [64,65], suggesting that exosomes can indirectly increase the genomic instability in the pre-cancer and cancer microenvironment, contributing to cancer initiation and heterogeneity.

Further, cancer-derived exosomes induce the formation of Web-like chromatin structures in neutrophils, named neutrophil extracellular traps, which are associated with a pro-thrombotic phenotype and the aggressiveness of the cancer [66,67].

Besides that, exosomes play a role in the transformation of normal cells to cancer cells [7,50,68]. This action is particularly regulated by the RNA content of the cancer-derived exosomes, which can be translated into proteins in the cytoplasm of recipient cells as demonstrated by Valadi et al. [69].

In this sense, Abd Elmageed et al. [70] showed that prostate cancer cell-derived exosomes are involved in tumor clonal expansion by reprogramming adipose-derived stem cells via trafficking of oncogenic transcripts (H-ras, K-ras, miR-125b, miR-130b, and miR-155). Supporting these data, Melo et al. [71] demonstrated that exosomes derived from cells and sera of breast cancer patients could promote the formation of tumors from nontumorigenic epithelial cells in a Dicer-dependent manner.

### 4.2. Cancer-Derived Exosomes Regulate Tumor Promotion and Progression

Although it is clear that cancer-driving mutations are necessary to its initiation, these mutations are not enough to promote its development [72,73].

Thus, cancer development requires sustaining proliferative signals to guarantee the clonal expansion of initiated cells, a step known as cancer promotion. In this sense, two pathways are commonly upregulated in most malignancies: activation of mitogen-activated protein kinase (MAPK) and phosphatidylinositol 3-kinase (PI3K)/Akt/mTOR [73].

In this sense, several studies have shown that cancer-derived exosomes can provide autocrine, paracrine, and endocrine signals, increasing the proliferation rate of non-cancer and cancer cells [74,75], contributing to both cancer promotion and progression [76,77].

In 2009, Qu et al. [78] reported that gastric cell line (SGC7901)-derived exosomes could promote the proliferation of gastric cancer cell lines (SGC7901 and BGC823) through the MAPK and PI3K/Akt/mTOR pathways, providing evidence that cancer-derived exosomes can regulate cancer growth. Supporting these data, in 2011, Kogure et al. [79] demonstrated that miRNAs present in hepatocellular carcinoma-derived exosomes could regulate transforming growth factor-beta activated kinase-1 (TAK-1), leading to hepatocellular cancer cell growth.

Besides promoting the upregulation of cell-cycle-related genes and increasing the S phase entry, cancer-derived exosomes can also downregulate the expression of cell cycle-arrest-related genes, contributing to the evasion of apoptosis. This is because esophageal adenocarcinoma-derived exosomes and microvesicles could promote the post-transcriptional downregulation of the phosphatase and tensin homolog (*PTEN*) and the apoptosis-inducing factor 2 (*AIFM2*) gene in a miR-25- and miR-210-dependent manner [80].

Moreover, exosomes of non-cancer cells, such as macrophages, could also promote cancer cell proliferation by different signaling pathways [77,81,82,83], reinforcing the crosstalk between the immune system and cancer development. This is because macrophage-derived exosomes play a key role in post-transcriptional control, regulating the phosphorylation of proteins in the recipient cells as revisited by Liu et al. [84]. Thus, both cancer- and non-cancer-derived exosomes can increase the intratumor heterogeneity, facilitating the gain and accumulation of passenger mutations during cancer progression [85,86].

### 4.3. Cancer-Derived Exosomes Regulate Several Steps of the Metastatic Process

#### 4.3.1. Cancer-Derived Exosomes as a Key Regulator of the Epithelial–Mesenchymal Transition (EMT)

Undoubtedly, metastasis is the most dramatic consequence of cancer, responsible for about 90% of cancer deaths globally [87].

Metastasis is a multistep process, which involves local invasion, intravasation, transport, extravasation, and colonization [88]. These steps require a series of genetic, biochemical, and morphological deregulations that are present in an evolutionarily conserved developmental program known as the epithelial–mesenchymal transition (EMT) [64,89,90,91].

The EMT is a natural process of transdifferentiation of epithelial cells to mesenchymal cells that is crucial for embryogenesis [92,93,94] and re-epithelization in tissue repair [95]. During embryogenesis, the EMT (EMT type I) gives rise to mesoderm (responsible for the formation of muscle, bone, and connective tissues) during gastrulation and neural crest delamination (which results in glial cell, adrenal gland, and epithelial pigmented cell formation) [90,96]. In adult life, the EMT plays a key role in tissue re-epithelization during wound healing (EMT type II) [95,97,98] but, when inappropriately active, such as occurs in carcinogenesis (EMT type III), the EMT causes important disturbances in epithelial tissue homeostasis and integrity, leading to cancer cell spread and metastasis [89,99].

The EMT (type III) is a consequence of cancer progression away from the cancer cells from the stroma, which is responsible for providing nutrients and oxygen support to the cells, creating a hypoxic environment. In addition, the partial reduction in the oxygen pressure leads to the activation of hypoxia-inducible factor 1 alpha (HIF-1α) in both cancer cells and cancer-associated fibroblasts (CAFs) [100,101,102].

HIF-1α nuclear translocation promotes the upregulation and stabilization of Snail and Twist, resulting in cadherin switching, which is characterized by the downregulation of E-cadherin (leading to a loss of intercellular adhesion and consequent activation of the Wnt/β-catenin pathway) and N-cadherin upregulation in cancer cells [103,104,105]. Combined with the F-actin reorganization of invadopodia sites, these actions create sites of transient adhesion that confer cell motility, facilitating the dissemination of cancer cells [89,106].

HIF-1α also acts as a key regulator of metabolic plasticity, promoting genetic and metabolic deregulations [90,107,108]. These deregulations drive the oxidative metabolism to glycolytic metabolism. This process is crucial to guaranteeing the energy supply (ATP) in hypoxic conditions [90]. In addition, glycolytic metabolism increases lactate production, which is generated as a byproduct of glycolysis.

L-Lactate is an important oncometabolite produced by the glycolytic cells within the TME, promoting a metabolic symbiosis between cancer cells and cancer-associated fibroblasts (CAFs) [109]. However, due to its high toxicity, L-lactate is transported out of the cytoplasm of CAFs to the extracellular compartment by a monocarboxylate transporter (MCT4), whose expression is upregulated by HIF-1α [110]. Thus, when released into the TME, the L-lactated CAFs can be uptaken by the MCT1 present in the plasma membrane of glycolytic cancer cells, which acts as a fuel source [111]. This is because cancer cells can oxidize the L-lactate to pyruvate in the mitochondria by lactate dehydrogenase, providing intermediate metabolites to the tricarboxylic acid cycle (TCA) [111,112].

However, the L-lactate exported to the extracellular space promotes the acidification of the TME [111]. The TME’s acidification inhibits the activation and proliferation of CD4+ and CD8+ lymphocytes, natural killer (NK) cells, and dendritic cells (DC) [111] as well as causes the polarization of the macrophages toward the M2 phenotype [111], contributing to immune evasion, which is recognized as a hallmark of cancer [113]. The TME’s acidification also induces the synthesis of metalloproteinases (MMPs) in both cancer and stromal cells, facilitating extracellular matrix (ECM) degradation and, therefore, cancer cell migration and spread [90,114].

Interestingly, studies have demonstrated that activation of HIF-1α by hypoxia increases the secretion of exosomes in both cancer [115,116,117,118] and non-cancer cells within the TME [119,120]. For this reason, hypoxia has been explored to increase the production of mesenchymal stem cell-derived exosomes for novel therapeutic strategies based on cell-free therapy [18,120,121]. This occurs because the hypoxia increases the L-lactate production and, therefore, reduces the pH, increasing the exosome release and uptake, contributing to the crosstalk between cancer and non-cancer cells within the TME [122,123,124].

In this sense, numerous studies have provided evidence that hypoxic cancer-derived exosomes regulate different EMT-related pathways in a miRNA-dependent manner [118,125,126]. In this context, it was reported that the miR-665 identified in hepatocellular carcinoma-derived exosomes can downregulate Hippo signaling through directly targeting tyrosine phosphatase receptor type B (PTPRB) [127], serving as a novel invasive biomarker for this malignancy [128]. This is because the Hippo tumor suppressor signaling pathway is crucial to controlling cell proliferation and apoptosis by inhibiting the oncogenic coactivators Yes-associated protein (YAP)/transcriptional coactivator with the PDZ-binding motif (TAZ) [129,130].

However, considering the plethora of biomolecules, especially miRNAs, delivered by cancer-derived exosomes, the mechanism of action of these vesicles on EMT could not be limited only to the Hippo signaling pathways.

In this sense, Yue et al. [131] showed that exosomal miR-301a, secreted by hypoxic glioblastoma cells, targets transcription elongation factor A like 7 (TCEAL7), leading to the activation of the Wnt/β-catenin signaling pathway, resulting in the expression of the EMT-related transcription factors Snail, Slug, and Twist. Similar results were verified by Nam et al. [132], who demonstrated that miR-301a functions as an oncogene in prostate cancer by directly targeting the p63 tumor suppressor, leading to loss of E-cadherin and EMT.

Thus, it is not surprising that cancer-derived exosomes can regulate different steps of the EMT, including cancer progression [133], dissemination [134,135], ECM remodeling [136,137], stemness [138], and metastasis [139], though different miRNAs.

Interestingly, studies have demonstrated that exosomes derived from cancer-associated macrophages can also regulate stem cells’ dormancy [140] and cell migration and invasion [141], providing evidence that exosomes are also implicated in metastasis.

In this sense, lung cancer cell-derived exosomes (from the A59 and H358 cell lines) alter the transcriptional and bioenergetic signature of M0 macrophages, leading them to an M2 phenotype [142]. However, the M2 macrophage-derived exosomes can transfer miR-21-5p and miR-155-5p to cancer cells, promoting the downregulation of transcription factor Brahma-related gene-1 (BRG1), leading to cell migration and invasion in colon cancer cells [141,143]. Gastric cancer showed similar results; M2 macrophage-derived exosome-mediated apolipoprotein E (ApoE) transfer was found to increase the cancer cell migration in a PI3K/Akt signaling pathway activation-dependent manner [144].

Altogether, these data reinforce the view that exosomes promote crosstalk between cancer and non-cancer cells within the TME, regulating the EMT and metastasis.

#### 4.3.2. Exosomes in Angiogenesis

Tumor vascularization is crucial to guaranteeing the support of nutrients and meeting oxygen needs to sustain cancer growth. For this reason, the activation of HIF-1α also serves as a signal to induce sustained angiogenesis [100,145]. Once phosphorylated, HIF-1α induces the expression of vascular endothelial growth factor (VEGF) [145,146,147,148]. VEGF binds to VEGF receptors (VEGFRs)-1, -2, and -3, which are expressed on vascular endothelial cells, regulating vessel formation through endothelial cell migration [149,150].

In this context, studies have demonstrated that cancer-derived exosomes act as a key regulator of angiogenesis [151,152]. This is because exosomes derived from cancer cells can stimulate endothelial cell migration and tube formation independently of uptake [153]. This response is mediated by the 189-amino-acid heparin-bound isoform of VEGF, which, unlike other common isoforms of VEGF, is preferentially enriched on the exosome surface [153].

However, cancer-derived exosomes can also promote angiogenesis in an uptake-dependent manner. In this sense, Li et al. [154] showed that hepatocellular carcinoma-derived exosomes transporting lysyl oxidase-like 4 (LOXL4) induce angiogenesis. In another study, Zhang et al. [155] demonstrated that ovarian cancer-derived exosomes expressing prokineticin receptor 1 (PKR1) promote angiogenesis by promoting the migration and tube formation of HUVEC cells. Similar results were also described by Umezu et al. [156], who demonstrated that hypoxia increases the production of multiple myeloma cell-derived exosomes transporting miR-135b, which can bind to factor-inhibiting hypoxia-inducible factor 1 (FIH-1) in endothelial cells, enhancing the formation of endothelial tubes. In another study, Zeng et al. [157] showed that colorectal cancer-derived exosomes drive miR-25-3p to endothelial cells, targeting Kruppel-like factors 1 and 4 (KLF2 and KF4, respectively) and promoting vascular permeability and angiogenesis.

Altogether, these data strongly suggest that cancer-derived exosomes are involved in angiogenesis.

#### 4.3.3. Cancer-Derived Exosomes Contribute to Pre-Metastatic Niche (PMN) Formation

Angiogenesis contributes to both cancer cell and cancer-derived exosome dissemination. However, the outcome of cancer metastasis depends on the interactions between metastatic cells and the host microenvironment [158]. These interactions between the cancer cells (“seeds”) and the host microenvironment (“soils”) were first discovered by the English surgeon Stephen Paget in 1889 [158]. About 40 years later (in 1928), James Ewing postulated that metastasis is determined by a mechanism associated with hemodynamic factors of the vascular system [159]. In a complementary hypothesis postulated in the 1970s, Isaiah Fidler demonstrated that, although the mechanical properties of blood flow are important, metastatic colonization only occurs at certain organ sites (organotropism) [159]. Fidler’s theory was supported by additional discoveries, which revealed that tumors induce the formation of microenvironments in distant organs, facilitating the survival and outgrowth of cancer cells before they arrived at these sites [159,160,161,162]. These predetermined microenvironments are termed ‘pre-metastatic niches’ (PMNs) [163].

In the context of the “seed and soil” theory (Paget’s theory), the exosomes are similar to fertilizers, which can make barren land fertile and facilitate the colonization of cancer cells [163,164,165,166]. This occurs because exosomes exhibit adhesion molecules on their surface, particularly integrins (ITGs), which bind to the ECM and organ-specific PMN receptors [164]. Supporting this theory, in a study evaluating the biodistribution of exosomes from different cancer cell lines, Hoshino et al. [167] provided evidence that cancer-derived exosomes are preferentially uptaken by tissues commonly recognized as metastatic sites. The authors also demonstrated that this site-specific biodistribution is associated with high expression levels of integrins (ITGα6, ITGβ4, and ITGβ1 for lung tropism; ITGβ5 and ITGαv for liver tropism; and ITGβ3 for brain tropism) [167], reinforcing the view that the integrins involved in PMN formation.

Cumulative studies have provided evidence that the local inflammatory microenvironment drives the formation of PMNs as revisited by Guo et al. [163]. In this sense, the exosomes play a key role in the metastatic process, inducing immune suppression in the PMN. This is because cancer cells release exosomes carrying programmed death-ligand 1 (PD-L1) [163]. When PD-L1 binds to programmed death receptor 1 (PD-1), which is mainly expressed on macrophages and activated T or B cells, it provides an inhibitory signal, inducing T cell apoptosis and/or inhibiting T cell activation and proliferation [168]. Thus, PD-L1/PD-1 binding allows the exosomes to circulate through the bloodstream without being recognized by immune cells [163,169,170].

In addition, cancer-derived exosomes contain many immunomodulatory molecules that can impair the immune cell function, resulting in an immunosuppressive pre-metastatic microenvironment [163]. These molecules can induce natural killer (NK) cell dysfunction, inhibit antigen-presenting cells, block T cell activation, and enhance apoptosis [171,172].

However, the effects of cancer-derived exosomes in PMN formation are not limited to immune suppression. Studies have demonstrated that exosomes released from hypoxic tumors increase angiogenesis and vascular permeability in the PMN by carrying different miRNAs, such as miR-105 and miR-25-3p, which can disrupt the vascular endothelial barrier by targeting specific gene products [166,167,173].

#### 4.3.4. Exosomes in Cancer Stem Cell (CSC) Formation

Cancer stem cells (CSCs), also known as tumor-initiating cells (TICs), are a subset of cancer cells that share various features with stem cells, including the ability to self-renew and differentiation into the heterogeneous lineages of cancer cells, producing a variety of tumor cell subpopulations [49,174,175,176]. In addition, these cells can induce cell cycle arrest (quiescent state), conferring chemo- and radio-resistance. This is because many common chemotherapeutic agents target the proliferating cells to lead to their apoptosis [174]. Furthermore, CSCs overexpress ATP-binding cassette (ABC) transporters, increasing chemotherapeutics’ efflux [177,178,179]. In addition, by exhibiting a high capability to repair DNA damage, the CSCs are resistant to radiation therapy (RT) [180,181]. Thus, although the origin of CSCs remains incompletely understood [182], it is clear that these cells are currently involved in therapeutic resistance [183].

Cumulative evidence has shown that genomic instability contributes to CSC formation and accelerates the development of many genetically variable cancer stem cells, increasing the intratumor heterogeneity [89,184,185,186,187].

However, recent studies have provided evidence that cancer-derived exosomes mediate crosstalk between the EMT and cancer stem cell (CSC) formation, acting as a key regulator of cell plasticity [49].

In this sense, numerous studies have shown that cancer-derived exosomes mediate the instability of cadherins (which was verified during the EMT) in recipient cells by transferring oncogenic microRNAs and long non-coding RNAs (lncRNAs) as revisited by Wang et al. [188].

The loss of E-cadherin, mediated by these non-coding RNAs [188], promotes β-catenin release into the cytoplasm [189]. Once translocated to the nucleus, β-catenin downregulates not only cell-junction-related genes (E-cadherin and claudin-7) [89,190] but also upregulates stemness-related genes, facilitating the formation of CSCs [191,192,193].

In addition, studies have also demonstrated that cancer-derived exosomes mediate drug resistance in several malignancies, which is considered a major impediment in medical oncology [194].

Basically, there are two main types of resistance in cancer: (i) inherent resistance, where insensitivity already exists before treatment; and (ii) acquired resistance, which subsequently appears following the initial positive response [194]. Interestingly, studies have demonstrated that cancer-derived exosomes mediate the acquired resistance by transferring microRNAs as revised by Bach et al. [194].

In this sense, Zheng et al. [195] showed that TME-derived exosomes transfer miR-21 to gastric cancer cells, resulting in therapeutic resistance to cisplatin. In another study, Richards et al. [196] provided evidence that CAF-derived exosomes confer resistance to gemcitabine on pancreatic ductal adenocarcinoma by transferring miR-146a.

Moreover, numerous studies have shown that CSC-derived exosomes transfer ATP-binding cassette (ABC), also known as multidrug resistance (MDR), proteins and mRNA, which are implicated in drug resistance [177,197,198], to recipient cells in different malignancies [199], such as breast cancer [200,201], prostate cancer [202], melanoma [203], and osteosarcoma [204], leading to drug-acquired resistance.

In addition, studies have also suggested that cancer-derived exosomes can confer resistance to radiotherapy by transferring circular RNA (circATP8B4) [205]. Further, Mustschelknaus et al. [206] showed that irradiated cancer cells increase the exosome uptake and improve the repair of DNA double-strand breaks.

## 5. Mesenchymal Stem Cell (MSC) Recruitment to the Tumor Microenvironment (TME)

Mesenchymal stem cells (MSCs) are important components of the tumor microenvironment (TME), which regulates and determines the final destination of cancer cells [207].

The inflammatory process creates an important network of communicability within the TME, acting as a mediator of the interaction between neoplastic and non-neoplastic cells through the production and secretion of a variety of pro-inflammatory cytokines, such as IL-1β, IL-6, IL-17, INF-γ, and TNF-α [208]. These pro-inflammatory cytokines, produced by the TME [209,210], recruit MSCs that naturally reside as pericytes in various tissues and (endogenous) organs [211] to the TME [212,213], driving cancer development and promoting changes in the tissue architecture [210]. Among these cytokines, IL-6 acts as a key component of the MSC recruitment [209], acting in a paracrine fashion on both endogenous and exogenous MSCs, stimulating the activation of the signal transducer and activator of transcription 3 (STAT3) and MAPK pathways, and enhancing the migratory potential and cell survival, which are necessary to MSC homing [209].

However, when naïve MSCs arrive at the TME, they are “educated” to have a pro-tumorigenic phenotype [214,215], supporting tumor growth through different mechanisms, such as: (i) differentiation in pro-tumorigenic stromal cells; (ii) suppression of the immune response; (iii) promotion of angiogenesis; (iv) enhancement of the EMT; (v) enrichment of CSCs; (vi) an increase in tumor cell survival; and (vii) promotion of cancer metastasis [214,216,217,218].

The role of MSCs in the TME is controversial since other studies have reported that MSCs elicit antitumorigenic potential by the: (i) enhancement of the immune response; (ii) inhibition of angiogenesis; (iii) regulation of cellular signaling; and (iv) induction of cell apoptosis [211,219,220,221,222].

Despite these controversial data, the tumor-suppressive effects are observed when MSCs are used in higher ratios than tumor cells [223]. Furthermore, the MSC function appears to be tissue-type-dependent and may rely on cancer education to reprogram a naïve MSC with antitumor effects [223]. For these reasons, efforts are mandatory to understand when MSCs promote or suppress carcinogenesis [224].

## 6. Mesenchymal Stem Cells as a Source of Exosomes for Cancer Treatment

In the last decade, MSCs have become the most used stem cell type for clinical applications. This is because these cells can easily be obtained from numerous adult and perinatal tissues, such as bone marrow, umbilical cord vein, Wharton’s jelly, adipose, and placental tissues, peripheral and menstrual blood, the liver, the spleen, and the pulp of deciduous teeth [16,225,226]. Furthermore, these cells can be propagated for several passages and show differential potential in various cell types and lineages, including adipose, osteogenic, and chondrogenic lineages (exogenous) [18,227,228]. Because of these advantages, these cells have been biotechnologically explored in advanced cellular therapies to treat numerous diseases [229,230,231].

For a long time, the therapeutic benefits of MSCs were associated with the replacement of dead cells [16,232]. However, cumulative evidence has demonstrated that less than 1% of transplanted MSCs survive for more than one week after systemic administration [225,232,233,234,235,236,237,238], suggesting that the therapeutic effects of MSCs are mediated by their “secretome” [226,239,240].

Supporting this hypothesis, several bioactive molecules identified in the MSCs’ secretome, such as chemokines, cytokines, interleukins, growth factors, lipid steroids, nucleotides, nucleic acids, ions, and metabolites [27,226], were already described to mediate biological functions [11,16,225,226,241] related to tissue regeneration [27,232,242].

These molecules can be found in free form or within exosomes [243]. However, whereas the soluble biomolecules present in the extracellular medium are subjected to rapid hydrolysis and/or oxidative effects, those present in exosomes are more stable [232]. This attracted the interest of researchers towards MSC-derived exosomes that could potentially be used in cell-free therapies [11,12,13].

Further, considering that MSCs can easily be manufactured on a large scale, these cells are an efficient mass producer of exosomes, allowing these vesicles to be used for therapeutic purposes [16,18].

Moreover, cell-free therapy possesses different advantages when compared with cell-based therapy, such as: (i) exosomes can be easily prepared and stored for a relatively long period without any toxic preservative, such as dimethylsulphoxide (DMSO); (ii) the use of exosomes instead of whole cells avoids possible complications associated with pulmonary embolism after intravenous infusion of MSCs; (iii) the use of exosomes avoids the risk of unlimited cell growth and tumor formation since exosomes do not divide; (iv) MSC-derived exosomes do not induce toxicity when repeatedly injected; (v) exosomes may be isolated from unmodified or genetically modified human MSCs; and (vi) the evaluation of a culture medium for safety and efficacy is much simpler to perform and analogous to that of conventional pharmaceutical agents [18,226,232,242,244,245].

All these advantages are directly related to the biological nature of the exosomes, which serve as intercellular messengers, delivering their contents to target cells. Moreover, exosomes exhibit a specific tropism for inflamed tissues, such as the TME [30].

Despite these advantages, the absence of scalable methods to isolate exosomes on a large scale has always been considered the main obstacle to the success of cell-free therapy. This is because most of the available technologies used for this purpose are time-consuming and generally provide few EVs [232]. However, improved methods for the isolation and purification of exosomes have facilitated the application of exosomes in clinical translation as previously discussed by us [18].

Thus, since the discovery that the therapeutic potential of MSCs is mediated by the exosomes produced and secreted by these cells, which have pleiotropic effects in recipient cells [246,247], including immunomodulatory properties [248,249], these vesicles became useful candidates for cancer treatment in a novel therapeutic approach known as cell-free therapy.

## 7. Clinical Applications of MSC-Derived Exosomes for Cancer Treatment

Considering that exosomes are natural nanocarriers of specific mRNAs, regulatory miRNAs and lncRNAs, and proteins, these vesicles have therapeutic potential for cancer in future clinical nanomedicine [250].

In this sense, recently, exosomes isolated from menstrual MSCs were found to inhibit tumor growth and angiogenesis of oral squamous cell carcinoma in a dose-dependent manner [251]. Supporting this antitumor effect, two independent studies showed that MSC-derived exosomes transporting TNF-related apoptosis-inducing ligand (TRAIL) induced apoptosis in 11 cancer cell lines in a dose-dependent manner [252,253].

In addition, MSC-derived exosomes can be engineered to act as vehicles for the delivery of specific miRNAs or chemotherapeutics, enlarging the range of therapeutic uses of these vesicles for cancer treatment [30]. In this sense, Lou et al. [254] demonstrated that exosomes derived from miR-122-transfected adipose tissue-derived MSCs increased the antitumor efficacy of sorafenib on hepatocellular carcinoma. Similar results were described by Li et al. [255], who demonstrated that exosomes derived from siGRP78-transfected bone marrow mesenchymal stem cells (BM-MSCs) suppress sorafenib resistance, inhibiting the growth and metastasis of hepatocellular carcinoma in vivo. Another study reported that exosomes derived from MSCs transfected with miR-199a reduce the proliferation, invasion, and migration of glioma cells via downregulation of ArfGAP with the GTPase domain, ankyrin repeat, and PH domain 2 (AGAP2) [256]. Similar results were also verified by Xu et al. [257], who demonstrated that BM-MSC-derived exosomes transporting miR-16-5p inhibit the proliferation, migration, and invasion and promote the apoptosis of colorectal cancer cells by downregulating ITGA2.

Using another biotechnological strategy, Melzer et al. [258] showed that taxol-loaded exosomes, obtained from continuously proliferating human MSC54 incubated with the drug (taxol), elicited anti-tumor effects in a mouse in vivo breast cancer model. In addition, the authors provided evidence that the intravenous injection of taxol-loaded MSC54 exosomes derived from the cell line displayed superior tumor-reducing capabilities compared with the application of taxol exosomes by oral gavage and that the exosome delivery route can affect the therapeutic efficacy of the cell-free therapy.

Studies on exosomes derived from different cells, including cancer cells, have also demonstrated that exosomes serve as an effective carrier of anti-tumor biomolecules and chemotherapeutic agents [259,260,261]. Based on this, in a study using cholangiocarcinoma cells, Ota et al. [262] demonstrated that exosome-encapsulated miR-30e, a widely studied tumor-suppressive miRNA [129,263,264], which negatively regulates tumor growth, invasion, and metastasis by targeting ITGB1, TUSC3, USP22, and SOX2 mRNAs [129,265,266,267,268], could suppress EMT in tumor cells by inhibiting Snail expression.

The antitumorigenic properties of MSC-derived exosomes have also attracted a great deal of interest due to the capability to drive specific molecules to cancer stem cells (CSCs) [208,269,270].

In this sense, Lee et al. [271] described that it is possible to reprogram CSCs into non-tumorigenic cells using osteogenic differentiating human adipose-derived exosomes (OD-EXOs) containing specific cargoes capable of inducing osteogenic differentiation of CSCs (alkaline phosphatase (ALPL), osteocalcin (BGLAP), and runt-related transcription factor 2 (RUNX2)). Furthermore, the authors demonstrated that the expression of ABC transporters, the breast cancer ge-e family (BCRA1 and BCRA2), and the ErbB gene family were significantly decreased in OD-EXO-treated CSCs, suggesting the exploration of MSC-derived exosomes for cancer therapy [271].

In an innovative approach, Tang et al. demonstrated that tumor cell-derived microparticles could be used as vectors to deliver chemotherapeutic drugs, resulting in cytotoxic effects and inhibition of drug efflux from cancer cells [259]. Similar results were later observed by Ma et al. [260], reinforcing the therapeutic use of exosomes for chemotherapeutic delivery to CSCs.

In another strategy, Kim et al. [272] developed an exosome-based formulation of paclitaxel (PTX), a commonly used chemotherapeutic agent, to overcome multidrug resistance (MDR) in cancer cells. For this, the authors employed three methods to incorporate PTX into exosomes: incubation at room temperature, electroporation, and mild sonication. Among these methods, electroporation resulted in the highest loading efficiency and sustained drug release [272]. However, the authors also showed that the PTX-loaded exosomes increased cytotoxicity by more than 50 times in drug-resistant MDCK_MRD1_ (Pgp+) cells [272]. Similar results were reported by Saari et al. [261], who described that prostate cancer-derived exosomes enhance the cytotoxicity of PTX in autologous cancer cells.

## 8. Future Prospects of Cell-Free Therapy for Cancer Treatment and Challenges to Be Overcome

Despite the numerous studies supporting the view that exosomes can be applied for cancer treatment in a new era of medicine, known as nanomedicine, there are considerable challenges to be solved, such as: (i) understanding the differences among exosomes from different sources to identify those whose content naturally elicits antitumor effects; and (ii) describing the mechanisms of action of these exosomes in order to explore their therapeutical potential for each histological type of cancer.

To overcome these difficulties, it is mandatory to develop novel in vitro methodologies that could provide detailed data about the exosomal biodistributions and provide information about the mechanisms of action of these vesicles, which is also required for the licensing of these exosomes as therapeutics by regulatory agencies. In this context, the future of exosomes as therapeutics for cancer depends on the improvement of 3D cell cultures.

Although 2D cultures are commonly used in cancer research [273], they do not recapitulate the complexity of the tumor microenvironment (TME). This is because 2D cultures do not exhibit the cell–ECM interactions or other cell types found within the TME, such as immune and stromal cells. Furthermore, despite the evidence that 2D cultures from cancer cell lines have a certain degree of heterogeneity [274], this heterogeneity does not reflect the genotypic and phenotypic cell heterogeneity verified within the TME [273,275]. Further, the cancer cell monolayer cannot recapitulate the biochemical properties, composition, tissue architecture, cell behavior, and exosome-mediated intercellular communication verified within the TME [276,277,278,279].

In this sense, 3D cell cultures and organoids have emerged as more reliable models to investigate the role of exosomes in cancer pathophysiology, allowing us to explore the mechanism of action of these vesicles in cancer biology [280].

Organoids are 3D cell culture systems formed through cell differentiation and the self-organization of pluripotent stem cells or tissue-derived progenitors that recapitulate the original function and structure of the tissue that they were derived from. Furthermore, these micro physiological systems can contain supporting stromal elements that mimic the TME [281].

Several studies suggest that the composition and dynamics of exosome secretion are influenced by many factors, including stimuli from the environment [282,283]. Presumably, extracellular vesicles derived from biomimetic tissue culture conditions better reflect secreted exosomes’ in vivo composition and function [25,280].

In this sense, Rocha et al. [283] compared the biochemical, transcriptomic, and proteomic profiles of exosomes from 2D and 3D cultures of gastric cancer cell line-derived exosomes. This study showed that 3D cultures produce more extracellular vesicles than 2D cultures. In addition, the global profile of microRNA and proteins was different compared with the 2D and 3D cultures, supporting the view that the tissue architecture affects the exosome content.

In another independent study, Thippabhotla et al. [280] analyzed RNA content from cervical cancer-derived extracellular vesicles (EVs) obtained from 2D and 3D cultures of HeLa cells and compared it with the RNA content from cervical cancer patient plasma-derived EVs. The study revealed remarkable differences between the EV content from the 2D and 3D cell cultures. Interestingly, the authors showed that the profile of small RNAs from a 3D-culture-derived EV exhibits a much higher similarity (~96%) to in vivo circulating EVs from cervical cancer patient plasma compared with a 2D-culture-derived EV. Similar results were verified by Villasante et al. [284], who showed that EVs derived from a 3D culture of Ewing’s sarcoma type 1 exhibit higher similarity to EVs derived from plasma patients than EVs from 2D cultures, supporting the view that these 3D culture models are better mimics of the TME, serving as powerful and useful models to study the role of exosomes in cancer biology and therapy.

In this context, different cancer organoids have been established as models to study different malignancies, including colorectal [285], colon [286], lung [287], liver [288], and pancreatic cancer [289]. In addition, these organoids have provided new clues about the exosome’s role in cancer pathophysiology and have enabled the description of the exosomal mechanism of action [290].

In this sense, using a 3D organoid model, Oszvald et al. [291] showed that fibroblast-derived EVs transporting amphiregulin (AREG) increase the number of proliferating colorectal cancer cells (CRC) in patient-derived organoid lines in an epidermal growth factor (EGF)-dependent manner. Further, although the authors observed that normal colon fibroblasts (NCF) activated with TGFβ (one of the most important activating factors of fibroblasts) secrete EVs with a different miRNA content profile compared with controls (NCF not active with TGFβ), they did not find differences in the biological effects between the EVs treated and not treated with TGFβ, suggesting that TGFβ-induced sorting of specific miRNAs into EVs does not play a major role in enhancing CRC proliferation [291]. Thus, the authors provided evidence that amphiregulin, transported by EVs, is a major factor in inducing CRC proliferation [291].

Despite the benefits of 3D cultures, to date, few works have studied the role of immobilized exosomes in the extracellular matrix of the TME. However, bioprinting technology has allowed the evaluation of the exosome effects on extracellular matrix remodeling [101,292,293,294]. This is because bioprinting technology is a powerful tool employed for tissue engineering, which allows for the precise placement of cells, biomaterials, and biomolecules in spatially predefined locales within confined 3D structures [295].

## 9. Conclusions

Exosomes are recognized as a key mediator of cell communication in both physiological and pathophysiological processes. For this reason, it is not surprising that these vesicles mediate cell-to-cell communication within the TME. In this sense, numerous studies have provided evidence that TME-derived exosomes are involved in all carcinogenesis steps, mediating crosstalk between cancer and non-cancer cells. This crosstalk not only increases the intratumor heterogeneity but recruits fibroblasts, pericytes, immune cells, and mesenchymal stem cells (MSCs) to the TME. When these cells enrich the TME, they can regulate the proteins, RNAs, and metabolites present in the cancer-derived exosomes. On the one hand, naïve MSCs can be polarized to type 2 MSCs (anti-inflammatory), which produce and secrete exosomes and cytokines that facilitate immune evasion; on the other hand, MSC-derived exosomes have emerged as useful candidates for cancer treatment in a novel therapeutic approach (cell-free therapy). This is because these vesicles can naturally deliver molecules able to suppress different steps of the carcinogenic process. Moreover, these vesicles can be biotechnologically engineered to be used to deliver drugs, especially cancer stem cells, which exhibit chemoresistance against multiple drugs. However, the therapeutic potential of these exosomes is conditioned to the MSC tissue since the exosomes share transcriptional and proteomic profiles similar to those of their producer cells. In this sense, novel efforts are needed to investigate the therapeutic potential of MSC-derived exosomes for different malignancies.

## Figures and Tables

**Figure 1 cells-10-02617-f001:**
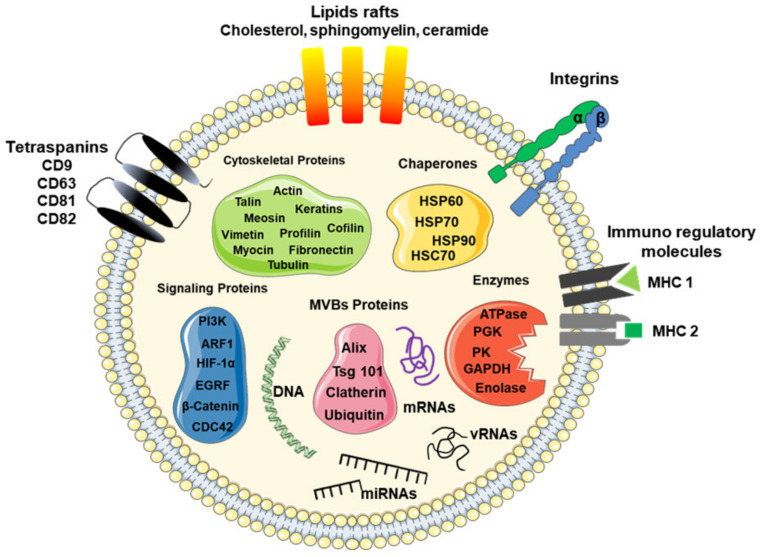
Schematic model of a typical exosome. The model shows a nanosized membrane-bound extracellular vesicle, with a diameter between 30 and 200 nm, expressing several proteins as a marker for exosomes, including tetraspanins (CD9, CD63, and CD81), Alix, Tsg101, and heat shock proteins (HSP-60, -70, and -90), as well as surface proteins, such as tetraspanins, integrins, immunoregulatory proteins (MHC-I and MHC-II), cytoskeletal proteins, signaling proteins, enzymes, and nucleic acids, such as coding RNAs (mRNAs) and non-coding RNAs (miRNAs and lncRNAs).

**Figure 2 cells-10-02617-f002:**
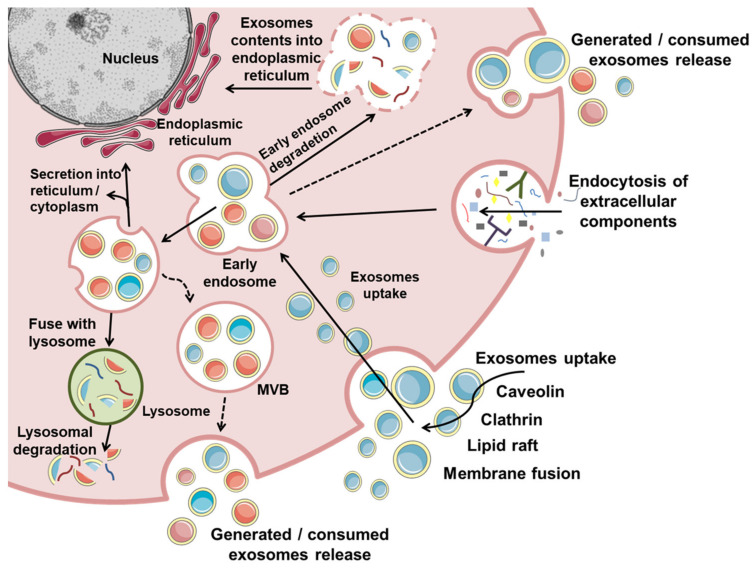
Biogenesis of exosomes: Exosome production requires double invagination of the plasma membrane, leading to intracellular multivesicular bodies (MVBs) containing intraluminal vesicles (ILVs). ILVs are ultimately secreted as exosomes with a size range of 30–200 nm in diameter through MVB fusion to the plasma membrane and exocytosis. The first membrane invagination forms a cup-shaped structure including cell-surface proteins and soluble proteins present in the extracellular environment. This process results in the formation of the early endosome. In this process, the trans-Golgi network and endoplasmic reticulum also contribute to the formation and content of the early endosome, which matures into late endosomes that eventually generate MVBs. MVBs form by inward invagination of the endosomal limiting membrane, resulting in MVBs containing several ILVs. MVBs can either fuse with lysosomes or autophagosomes to be degraded or fuse with the plasma membrane to release the contained ILVs as exosomes. Exosome uptake: Exosomes are continuously being generated by and taken up by cells. Exosomes that are taken up can be degraded by lysosomes. In contrast, exosomes that enter cells may fuse with the pre-existing early endosome and subsequently disintegrate and release their content into the cytoplasm. Alternatively, endosomes can fuse back with the plasma membrane, releasing exosomes outside the cells.

**Figure 3 cells-10-02617-f003:**
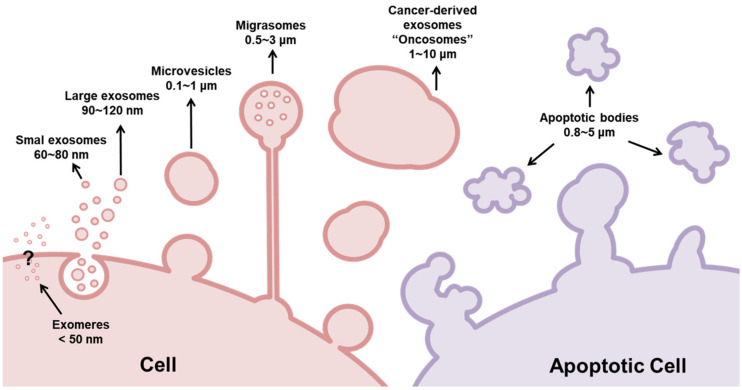
Classification of extracellular vesicles (EVs) according to their size. Basically, EVs are classified as exosomes (30–150 nm), microvesicles (100–1000 nm), and apoptotic bodies (800–5000 nm). While microvesicles and exosomes can operate as ‘safe containers’ mediating intercellular communication, apoptotic bodies appear after the disassembly of an apoptotic cell into subcellular fragments. Although they were previously regarded as garbage bags, emerging evidence supports the view that the apoptotic bodies are capable of delivering useful materials to healthy recipient cells. Different from exosomes, microvesicles are generated from the direct outward blebbing and pinching of the plasma membrane. Similar to exosomes, these vesicles carry proteins, nucleic acids, and bioactive lipids to recipient cells; however, they are larger than exosomes. Exosomes are conserved structures that originate as intraluminal vesicles during the assembly of multivesicular bodies, mediating cell-to-cell communication. However, current studies show that cancer-derived exosomes are larger than those secreted by normal/healthy cells. For this reason, these nanosized EVs were subclassified as exomers (<50 nm), small exosomes (60–80 nm), large exosomes (90–120 nm), and oncosomes (100–10,000 nm). Recently, a novel type of EV was described: migrasomes (500–3000 nm). Migrasomes are vesicular structures that mediate migracytocis, a cell migration mechanism mediated by these EVs.

## Data Availability

Not applicable.

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
