# Peer review of "Exosomes in the Tumor Microenvironment: From Biology to Clinical Applications"

_cells, 2021, doi:10.3390/cells10102617_

Round 1
Reviewer 1 Report
The review by Rodrigues da Costa et al examines the role of cancer-derived exosomes in tumor microenvironment and in cancer biology with a focus on the role of CSCs. The outline of the manuscript is acceptable , but I strongly suggest to make some corrections as reported below, and to use a native English speaker to proofread the manuscript. Too many language mistakes are present throughout the text.
Specific comments:
- Figure 2 presents some typos such as exossomes, lyssosomes etc etc..I suggest to increase the size of characters, and to remove some descriptions in the left part of the figure.
The right part of the figure 2 is a repetition of fig 1, therefore i suggest to remove it.
- The concept that exosomes derived from cancer cells are larger than those derived from non cancer cells is unknown to me (line 178-179). If so, please add the reference in the revised text, or remove this sentence. What is known is that the size of EVs, in general, refers to the biogenesis of EVs, and larger EVs derive from plasma membrane shedding in cancer cells. In support of this, the description of all subsets of EVs recoverable extracellularly, including exomeres and small exosomes (Figure 3), derives from a study conducted by Zhang H et al on tumor cell line ( B16-F10) (doi.org/10.1038/s41556-018-0040-4). Please add this reference in the text and rephrase accordingly this part in section 4.
- Line 183 add “not” between is and unanimous
- Line 320 –the references indicated regarding the role of HIF-1a and CAFs are inapproriate. Please add Petrova V 2018 (doi.org/10.1038/s41389-017-0011-9)
- I strongly recommend to create a Table listing molecules type, biological functions, origin (soluble or exosomal origin) isolated from MSCs
- Line 657- remove the word “ role”
- Line 662 -change mimetic with mimic
- Line 707- remove “nor only”
- A proof reading by a native speaker English is strongly recommended. Here some example of common mistakes found through out the text:
Line 117 change producing with production
Line 118 correct lead with leads
Line 127 correct releases with release
Line 156 remove been
Line 159 correct evidence with evidences and remove been
Line 219 remove been
Line 228 remove been etc. etc etc etc
Author Response
Response to reviewer 1 comments
We thank for the comments and suggestion. As suggested,
Point 1: Figure 2 presents some typos such as exossomes, lyssosomes etc etc..I suggest to increase the size of characters, and to remove some descriptions in the left part of the figure. The right part of the figure 2 is a repetition of fig 1, therefore i suggest to remove it.
Response 1: We corrected Figure 2 as it was suggested. We increased the size of characters and removed the right part of figure 2.
Point 2: The concept that exosomes derived from cancer cells are larger than those derived from non cancer cells is unknown to me (line 178-179). If so, please add the reference in the revised text, or remove this sentence. What is known is that the size of EVs, in general, refers to the biogenesis of EVs, and larger EVs derive from plasma membrane shedding in cancer cells. In support of this, the description of all subsets of EVs recoverable extracellularly, including exomeres and small exosomes (Figure 3), derives from a study conducted by Zhang H et al on tumor cell line ( B16-F10) (doi.org/10.1038/s41556-018-0040-4). Please add this reference in the text and rephrase accordingly this part in section 4.
Response 2: The concept that cancer cell-derived exosomes are larger than those derived from noncancer cells is well stated in the literature, as discussed in different published studies and reviews (described above). This occurs because cancer cells overexpress many abnormal and transforming macromolecules, including oncoproteins. Thus, large exosomes carrying these abnormal proteins and exhibiting a diameter between 100-400 nm are classified as oncosomes, whereas those produced by the plasma membrane blebbing from cancer cells and, induced by the silencing of DIAPH3, by overexpression of MyrAKt1, HB-EGF, and caveolin-1, or by the activation of EGFR, and exhibiting a diameter between 100-1000 nm are classified as large oncosomes. To clarify this concept, additional information was added in lines 186-192.
Minciacchi V, et al. MYC Mediates Large Oncosome-Induced Fibroblast Reprogramming in Prostate Cancer. Microenrionment and Immunology. 2017, 77:9, 2306-2317 doi: 10.1158/0008-5427.CAN-16-2942
Minciacchi V, et al. Large oncosomes contain distinct protein cargo and represent a separate function class of tumor-derived extracellular vesicles. Oncotarget, 2015, 6:13, 11327-11341
Meehan, B, et al. Oncosomes- large and small: what are they, where they came from? JEV, 2016, 5:33109 doi:10.3402/jev.v5.33109
Syn N, et al. Exosome-Mediated metastasis: from epithelial-mesenchymal transition to escape from immunosurveillance. Trends in Pharmacological Sciences. Doi:10.1016/j.tips.2016.04.006
Kahlert C, et al. Exosomes in tumor microenvironment influence cancer progression and metastasis. J Mol Med. 2013, 91, 431-437 doi:10.1007/s00109-013-1020-6
Chalmin F, et al. Membrane-associated HSP72 from tumor-derived exosomes mediates STAT3-dependent immunosuppressive function of mouse and human myeloid derived suppressor cells. J Clinical Investigations. 2010, 1202, 457-471 doi:10.1172/JCI40483
Poggio M, et al. Suppression of exosomal PD-L1 induces systemic anti-tumor immunity and memory. Cell. 2019, 177:2, 414-427 doi:10.1016/j.cell.2019.02.016
Atay S, et al. Tumor-derived exosomes: A message delivery system for tumor progression. Communicative and integrative Biology. 2014, 7:1, e28231 doi:10.4161/cib.28231
Ponit 3: Line 183 add “not” between is and unanimous
Response 3: We apologize for the lack of “not” in line 183. So, the sentence was corrected for “there is no unanimous”
Point 4: Line 320 –the references indicated regarding the role of HIF-1a and CAFs are inapproriate. Please add Petrova V 2018 (doi.org/10.1038/s41389-017-0011-9)
Response 4: As suggested, the reference cited in line 320 was changed to Petrova (2018).
Point 5: I strongly recommend to create a Table listing molecules type, biological functions, origin (soluble or exosomal origin) isolated from MSCs
Response 5: We agree that a table listing the repertoire of biomolecules identified in MSC and their biological functions could be beneficial. However, there are different databases allowing access to such information. Moreover, there are many different molecules, including coding and non-coding RNA, proteins, lipids, and metabolites identified in MSCs, which can differ according to the origin of MSC. For this reason, we added some databases helping to access the MSC biomolecules list.
Point 6: Line 657- remove the word “ role”
Response 6: The word “role” in line 657 was removed, as suggested
Point 7: Line 662 -change mimetic with mimic
Response 7: The word “mimic” in line 662 was changed to “mimic”, as suggested
Point 8: Line 707- remove “nor only”
Response 8: The term “not Only” was removed
Point 9: A proof reading by a native speaker English is strongly recommended. Here some examples of common mistakes found through out the text:
Response 9: The English was revised by a native speaker and the mistakes were corrected
Point 10: Line 117 change producing with production
Response 10: The word “producing” was replaced by “production”
Point 11: Line 118 correct lead with leads
Response 11: The word “lead” was replaced by “leads”
Point 12: Line 127 correct releases with release
Response 12: The word “releases” in line 127 was corrected to “release”
Point 13: Line 156 remove been
Response 13:The word “been” was removed
Point 14: Line 159 correct evidence with evidences and remove been
Response 14: The word “evidence” was corrected to “evidences” and the word “been” was removed
Point 15: Line 219 remove been
Response 15: The word “been” was removed
Point 16: Line 228 remove been etc. etc etc etc
Response 16: The word “been” was removed

Reviewer 2 Report
Costa et. al. have written a very well surveyed review on "exosomes in the tumor microenvironments" and I enjoyed reading it.
- A minor suggestion for the authors would be to include in section 4.3.3 some of the first reports (mentioned below) on tumor exosome-mediated immune suppression.
- Theodoraki, M. N., Yerneni, S. S., Hoffmann, T. K., Gooding, W. E., & Whiteside, T. L. (2018). Clinical significance of PD-L1+ exosomes in plasma of head and neck cancer patients. Clinical cancer research, 24(4), 896-905.
- Hong, C. S., Sharma, P., Yerneni, S. S., Simms, P., Jackson, E. K., Whiteside, T. L., & Boyiadzis, M. (2017). Circulating exosomes carrying an immunosuppressive cargo interfere with cellular immunotherapy in acute myeloid leukemia. Scientific reports, 7(1), 1-10.
- In section 8, the authors should also mention that in the tumor microenvironment, exosomes are present not just in soluble form but also in an immobilized form within the extracellular matrix (ECM). However, to date, most experiments with tumor exosomes are performed by utilizing soluble exosomes alone, and very few studies involve immobilized exosome microenvironments. This is partly because in vitro models that recapitulate such environments are limited. One toolset available is to use bioprinting technology to create exosome-ECM microenvironments (reference below)
Chen, G., Huang, A. C., Zhang, W., Zhang, G., Wu, M., Xu, W., ... & Guo, W. (2018). Exosomal PD-L1 contributes to immunosuppression and is associated with anti-PD-1 response. Nature, 560(7718), 382-386.
Girigoswami, K., Saini, D., & Girigoswami, A. (2020). Extracellular matrix remodeling and development of cancer. Stem Cell Reviews and Reports, 1-9.
Huleihel, L., Hussey, G. S., Naranjo, J. D., Zhang, L., Dziki, J. L., Turner, N. J., ... & Badylak, S. F. (2016). Matrix-bound nanovesicles within ECM bioscaffolds. Science advances, 2(6), e1600502.
Yerneni, S. S., Whiteside, T. L., Weiss, L. E., & Campbell, P. G. (2019). Bioprinting exosome-like extracellular vesicle microenvironments. Bioprinting, 13, e00041.
Author Response
Response to reviewer 2 comments
Point 1: A minor suggestion for the authors would be to include in section 4.3.3 some of the first reports (mentioned below) on tumor exosome-mediated immune suppression.
Theodoraki, M. N., Yerneni, S. S., Hoffmann, T. K., Gooding, W. E., & Whiteside, T. L. (2018). Clinical significance of PD-L1+ exosomes in plasma of head and neck cancer patients. Clinical cancer research, 24(4), 896-905.
Hong, C. S., Sharma, P., Yerneni, S. S., Simms, P., Jackson, E. K., Whiteside, T. L., & Boyiadzis, M. (2017). Circulating exosomes carrying an immunosuppressive cargo interfere with cellular immunotherapy in acute myeloid leukemia. Scientific reports, 7(1), 1-10.
Response 1: We thank for the recommendation to include the reports mentioned. The references were added (in lines 439-440).
Point 2: In section 8, the authors should also mention that in the tumor microenvironment, exosomes are present not just in soluble form but also in an immobilized form within the extracellular matrix (ECM). However, to date, most experiments with tumor exosomes are performed by utilizing soluble exosomes alone, and very few studies involve immobilized exosome microenvironments. This is partly because in vitro models that recapitulate such environments are limited. One toolset available is to use bioprinting technology to create exosome-ECM microenvironments (reference below)
Chen, G., Huang, A. C., Zhang, W., Zhang, G., Wu, M., Xu, W., ... & Guo, W. (2018). Exosomal PD-L1 contributes to immunosuppression and is associated with anti-PD-1 esponse. Nature, 560 (7718), 382-386.Girigoswami, K., Saini, D., & Girigoswami, A. (2020). Extracellular matrix remodeling and development of cancer. Stem Cell Reviews and Reports, 1-9.Huleihel, L., Hussey, G. S., Naranjo, J. D., Zhang, L., Dziki, J. L., Turner, N. J., ... & Badylak, S. F. (2016). Matrix-bound nanovesicles within ECM bioscaffolds. Science advances, 2 (6), e1600502.Yerneni, S. S., Whiteside, T. L., Weiss, L. E., & Campbell, P. G. (2019). Bioprinting exosome-like extracellular vesicle microenvironments. Bioprinting, 13, e00041.
Response 2: We thank you for the suggestion to discuss the limits of 2D and 3D cultures as a model to study the role of immobilized exosomes and the applicability of bioprinting technology as a powerful tool to understand the exosome role in ECM remodeling. For this reason, we added a novel paragraph about this issue, citing the references suggested by the reviewer (lines 717-722).
Reviewer 3 Report
This review deals with exosomes in the tumor microenvironment. There are some nice illustrations, but there are already quite a few such reviews in this field, and as such I think there needs to be significant work in order to be publishable.
- The language needs extensive work throughout. There are grammatical and spelling issues throughout. In the abstract alone, "Cancer is one of the most important health problems", "cancer heterogeneity remains challenging the therapeutics", "CSCs, which exhibit resistance".
- There needs to be more clarity of some details. This may be due to unclear language. For example Line 179 - "the size difference can be attributed to the heterogeneous nature of cancer cells"? How so?
- Line 78 - it is arguable that vesicles were described earlier than the late 1980's. It is worth linking to some other historical reviews on the topic of EVs.
- There is repetition of the aims of the review. Authors should just set this out at the start.
- Spelling: e.g. oncossomes (figure 3), lysyl (409), of (633).
- I think the authors need to clarify that EMT is a component of metastasis, as it reads as though they are the same process (304-5), and interacts with processes such as ECM remodelling etc (377-8).
- The section on pre-metastatic niche (4.3.3) is weak. There is a lot of work in this field, this section barely touches the surface.
Overall I think this review needs work to highlight its novelty in order to stand out in the field.
Author Response
Response to reviewer 3 comments
Point 1: A minor suggestion for the authors would be to include in section 4.3.3 some of the first reports (mentioned below) on tumor exosome-mediated immune suppression.
Theodoraki, M. N., Yerneni, S. S., Hoffmann, T. K., Gooding, W. E., & Whiteside, T. L. (2018). Clinical significance of PD-L1+ exosomes in plasma of head and neck cancer patients. Clinical cancer research, 24(4), 896-905.
Hong, C. S., Sharma, P., Yerneni, S. S., Simms, P., Jackson, E. K., Whiteside, T. L., & Boyiadzis, M. (2017). Circulating exosomes carrying an immunosuppressive cargo interfere with cellular immunotherapy in acute myeloid leukemia. Scientific reports, 7(1), 1-10.
Response 1: We thank for the recommendation to include the reports mentioned. The references were added (in lines 439-440).
Point 2: In section 8, the authors should also mention that in the tumor microenvironment, exosomes are present not just in soluble form but also in an immobilized form within the extracellular matrix (ECM). However, to date, most experiments with tumor exosomes are performed by utilizing soluble exosomes alone, and very few studies involve immobilized exosome microenvironments. This is partly because in vitro models that recapitulate such environments are limited. One toolset available is to use bioprinting technology to create exosome-ECM microenvironments (reference below)
Chen, G., Huang, A. C., Zhang, W., Zhang, G., Wu, M., Xu, W., ... & Guo, W. (2018). Exosomal PD-L1 contributes to immunosuppression and is associated with anti-PD-1 esponse. Nature, 560 (7718), 382-386.Girigoswami, K., Saini, D., & Girigoswami, A. (2020). Extracellular matrix remodeling and development of cancer. Stem Cell Reviews and Reports, 1-9.Huleihel, L., Hussey, G. S., Naranjo, J. D., Zhang, L., Dziki, J. L., Turner, N. J., ... & Badylak, S. F. (2016). Matrix-bound nanovesicles within ECM bioscaffolds. Science advances, 2 (6), e1600502.Yerneni, S. S., Whiteside, T. L., Weiss, L. E., & Campbell, P. G. (2019). Bioprinting exosome-like extracellular vesicle microenvironments. Bioprinting, 13, e00041.
Response 2: We thank you for the suggestion to discuss the limits of 2D and 3D cultures as a model to study the role of immobilized exosomes and the applicability of bioprinting technology as a powerful tool to understand the exosome role in ECM remodeling. For this reason, we added a novel paragraph about this issue, citing the references suggested by the reviewer (lines 717-722).
Point 3: The language needs extensive work throughout. There are grammatical and spelling issues throughout. In the abstract alone, "Cancer is one of the most important health problems", "cancer heterogeneity remains challenging the therapeutics", "CSCs, which exhibit resistance".
Response 3: We inform that the English was reviewed.
Point 4: There needs to be more clarity of some details. This may be due to unclear language. For example Line 179 - "the size difference can be attributed to the heterogeneous nature of cancer cells"? How so?
Response 4: In general, the extracellular vesicles, including exosomes, are heterogeneous in terms of size. However, it is well stated that by overexpressing many oncogenes, the cancer-derived exosomes are larger than those produced by non-cancer cells. Aiming to clarify this point, we added two novel sentences (lines 188-193) describing that oncosomes and large oncosomes carrying abnormal and transforming macromolecules.
Point 5: Line 78 - it is arguable that vesicles were described earlier than the late 1980's. It is worth linking to some other historical reviews on the topic of EVs.
Response 5: Exosomes were first described in two independent studies published in 1938 by Harding et al. and Pan and Johnstone (references added in line 79). To clarify that this date refers to the discovery of exosomes, we changed the lines 79-82.
Point 6: There is repetition of the aims of the review. Authors should just set this out at the start.
Response 6: This repetition was removed. We apologize for this.
Point 7: Spelling: e.g. oncossomes (figure 3), lysyl (409), of (633).
Response 7: We apologize for the typo “oncossomes” (figure 3), as well as by the words “lysyl” and “of” Both mistakes were corrected and the figure was replaced.
Point 8: I think the authors need to clarify that EMT is a component of metastasis, as it reads as though they are the same process (304-5), and interacts with processes such as ECM remodeling etc (377-8).
Response 8: Aiming to clarify that the EMT is a key component of metastasis, we rewrite the paragraph, describing that “The metastatic is a multistep process, which involves local invasion, intravasation, transport, extravasation, and colonization [88]. These steps require a series of genetic, biochemical and morphological deregulations which are present in an evolutionarily conserved developmental program known as epithelial-mesenchymal transition (EMT)” (lines 318-322).
Point 9: The section on pre-metastatic niche (4.3.3) is weak. There is a lot of work in this field, this section barely touches the surface.
Response 9: Aiming to provide more details about the exosome role in pre-metastatic niche formation, we rewrite the section 4.3.3